# Pharmaceutical Approaches to Normal Tension Glaucoma

**DOI:** 10.3390/ph16081172

**Published:** 2023-08-17

**Authors:** Maria Letizia Salvetat, Francesco Pellegrini, Leopoldo Spadea, Carlo Salati, Marco Zeppieri

**Affiliations:** 1Department of Ophthalmology, Azienda Sanitaria Friuli Occidentale, 33170 Pordenone, Italy; 2Eye Clinic, Policlinico Umberto I, “Sapienza” University of Rome, 00142 Rome, Italy; 3Department of Ophthalmology, University Hospital of Udine, 33100 Udine, Italy

**Keywords:** normal tension glaucoma, intraocular pressure, medical therapy, glaucomatous optic neuropathy, visual field defects, target IOP, neuroprotection

## Abstract

Normal tension glaucoma (NTG) is defined as a subtype of primary open-angle glaucoma (POAG) in which the intraocular pressure (IOP) values are constantly within the statistically normal range without treatment and represents approximately the 30–40% of all glaucomatous cases. The pathophysiology of this condition is multifactorial and is still not completely well known. Several theories have been proposed to explain the onset and progression of this disease, which can be divided into IOP-dependent and IOP-independent factors, suggesting different therapeutic strategies. The current literature strongly supports the fundamental role of IOP in NTG. The gold standard treatment for NTG tends to be based on the lowering IOP even if “statistically normal”. Numerous studies have shown, however, that the IOP reduction alone is not enough to slow down or stop the disease progression in all cases, suggesting that other IOP-independent risk factors may contribute to the NTG pathogenesis. In addition to IOP-lowering strategies, several different therapeutic approaches for NTG have been proposed, based on vaso-active, antioxidant, anti-inflammatory and/or neuroprotective substances. To date, unfortunately, there are no standardized or proven treatment alternatives for NTG when compared to traditional IOP reduction treatment regimes. The efficacy of the IOP-independent strategies in decreasing the risk or treating NTG still remains inconclusive. The aim of this review is to highlight strategies reported in the current literature to treat NTG. The paper also describes the challenges in finding appropriate and pertinent treatments for this potentially vision-threatening disease. Further comprehension of NTG pathophysiology can help clinicians determine when to use IOP-lowering treatments alone and when to consider additional or alternatively individualized therapies focused on particular risk factors, on a case-by-case basis.

## 1. Introduction

### 1.1. Normal Tension Glaucoma (NTG) Definition 

Glaucoma is a multifactorial, chronic, progressive and degenerative optic neuropathy (ON) characterized by the loss of retinal ganglion cell (RGC) bodies and axons, which results in thinning of the retinal nerve fiber layer (RNFL), typical optic nerve head (ONH) cupping and correspondent specific and irreversible visual field (VF) defects, with consequent visual disability and a reduction in quality of life [1]. Glaucomatous optic neuropathy (GON) represents the most common ON affecting humans, and it is the leading cause of irreversible blindness worldwide. Its prevalence is estimated to extremely increase in the near future [2].

Glaucoma represents a group of heterogeneous diseases. Primary open-angle glaucoma (POAG) is the most prevalent form in Western countries and it is characterized by GON, correspondent typical VF defects, open anterior chamber angle on gonioscopy and intraocular pressure (IOP) values higher than the statistically normal range, i.e., >21 mmHg [3]. Normal tension glaucoma (NTG), first recognized as a clinical entity by von Graefe in 1857 [4], is a subtype of primary glaucoma characterized by open-angle and IOP values constantly within the statistically normal range without treatment [5]. The definitions of NTG, however, may vary slightly amongst different countries. The European Glaucoma Society (EGS) Guidelines, published in 2021 [3], state that “NTG is a specific type of POAG characterized by glaucomatous optic nerve head damage and corresponding visual field defects in patients with IOP consistently less than 21 mmHg”. The Preferred Practice Pattern Guidelines published in 2021 by the American Academy of Ophthalmology (AAO) define the NTG as “a common form of POAG, i.e., a chronic, progressive optic neuropathy that results in a characteristic optic nerve head cupping, retinal nerve fiber layer thinning and functional visual field loss, in which there is no measured elevation of the IOP” [6]. 

In 2015, the Canadian Ophthalmological Society Guidelines reported NTG as “a subgroup of POAG with characteristic visual field defects and glaucomatous optic nerve head changes in patients having normal IOP levels less than 21 mmHg” [7]. The Asia-Pacific Glaucoma Guidelines, published in 2016, reported that “the NTG is a condition in which the typical glaucomatous progressive optic nerve damage and visual field loss occur although the intraocular pressure remains normal” [8]. The Japanese Glaucoma Society (JGS) published guidelines in 2023, which defined NTG as “a subtype of POAG in which the IOP always remains within the statistically determined normal range during the developmental process of GON” [9].

### 1.2. Epidemiology of NTG

Epidemiological studies have reported that the NTG represents about 20–40% of all cases of POAG amongst Caucasians or Africans [10], whereas, for unclear reasons, Asian populations, especially Korean, Japanese and Chinese subjects, appear to be especially susceptible to NTG, with a prevalence ranging between 47% and 92% of all POAG patients [11]. 

### 1.3. Diagnosis, Differential Diagnosis and Clinical Features of NTG

Considering that the screening modalities for glaucoma are based on the IOP measurements, the diagnosis of NTG is understandably particularly difficult and often delayed. In the suspicion of NTG, two fundamental questions should be answered:


(1) Is it really a glaucoma?


NTG diagnosis requires the exclusion of other causes of non-glaucomatous ONH cupping [12,13,14], which includes the following:oCongenital disorders: coloboma of the OHN, pits, ONH oblique insertion and autosomal dominant optic atrophy (Kjer type); oAcquired disorders: history of steroid use, previous trauma or surgery that may induce prior elevated IOP, hemodynamic crisis, optic neuritis, anterior optic ischemic neuropathy (both arteritic and non-arteritic), compressive lesions of the ONH and optic tract (meningioma, pituitary adenoma, craniopharyngioma, internal carotid artery aneurysm, etc.). 

Glaucoma and non-arteritic anterior optic ischemic neuropathy (NA-AION) are the most common causes of irreversible optic neuropathy in adults; even if the clinical features of NA-AION in the acute phase are typical of this condition. NA-AION may present with ONH cupping in the non-acute phase, simulating a GON. Although having similar patterns of RNFL loss and VF defect, NTG and NA-AION have different ONH microvasculature characteristics as shown by the optical coherence tomography angiography, suggesting different mechanisms of vascular impairment [15]. 

A correct NTG diagnosis may require a complete neuro-ophthalmological examination and neuroimaging in the presence of specific features that may increase the likelihood of identifying an intracranial expansive lesion, which include younger age, lower visual acuity levels, highly asymmetric cupping between fellow eyes, vertically aligned VF defects, neuro-retinal rim pallor and disease progression despite the optimization of the known risk factors for NTG [16]. Several authors have suggested that NTG could be a non-glaucomatous disease, representing a group of disorders with GON as a characteristic clinical feature [17].


(2) Is it really a glaucoma with “normal IOP”?


The diagnosis of NTG requires the exclusion of all types of secondary glaucoma, which also must include the potential misclassification of POAG. The differential diagnosis between NTG and POAG is based on the exclusion of elevated IOP values or IOP spikes that cannot always be detected with the currently available tonometric methods that measure IOP in a static manner. 

Several factors can lead to an underestimation of the IOP that can lead to a misclassification of patients as NTG instead of POAG, which include:Diurnal, nocturnal and postural IOP variations, with the possibility of missing peaks of elevated IOP during non-office hours, especially during nocturnal and sleeping periods [18].Corneal thickness variations, in which thin corneas can cause an underestimation of the IOP measured with several tonometers, including the gold standard Goldmann applanation tonometry (GAT) [19].Corneal biomechanics variations, with more deformable corneas inducing an underestimation of the IOP measured with various tonometers, including GAT [20].

Although being classified as a POAG subtype, aside from the IOP values, NTG patients are characterized by typical clinical features that can help in the differentiation from POAG [5,17,21], which include: Older age and higher prevalence of females, Asian population and high myopia;Neuro-retinal rim damage prevalently situated in the inferotemporal quadrant;Narrower neuro-retinal rim for a given VF defect amount;More frequent disc hemorrhages, focal defects of the lamina cribrosa and beta peri-papillary zone;More frequent focal areas of cupping at the disc margin;VF defects closer to the fixation point, deeper and more focal;More frequent association with a variety of systemic diseases that can induce an ischemic and/or hypoxic damage of the ONH, including migraine, Raynaud’s phenomenon, primary vascular dysfunction (Flammer syndrome), systemic hypotension (especially nocturnal arterial hypotension) and obstructive sleep apnea syndrome (OSAS). A systemic evaluation for potentially contributing conditions, such as OSAS or Raynaud’s phenomenon or systemic hypotension, is important, especially in cases of disease progression refractory to IOP-lowering therapy.

## 2. Pathophysiology of the NTG as Rationale for Different Therapeutic Strategies

The comprehension of the genesis of a specific disease is of central importance for the choice of the correct therapeutic strategy. Unfortunately, the pathophysiology of glaucoma, especially that of NTG, is complex and not fully understood yet. The first event in the onset of glaucoma is believed to be the damage of the RGC axons, in which a Wallerian-type degeneration results in the characteristic glaucomatous ONH cupping. The death of RGC bodies has been demonstrated to occur over hours to days upon the onset of axon degeneration [22]. Studies have suggested that the time interval between axons degeneration and cell body death could increase the opportunities for the RGCs’ survival if targeted therapies are adopted in due time [23,24,25]. 

A programmed cell self-killing mechanism, known as apoptosis, seems to be the dominant mechanism of RGC death in glaucoma [22,26,27]. Once the apoptotic process has started, the damaged RGCs can induce the degeneration of adjacent cells through the so-called trans-synaptic degeneration process, which seems to be independent of the primary insult [22,26,27]. 

As known, mammalian RGCs are not able to regenerate, and their death is considered an irreversible event that can only be prevented, not treated [22]. Considering that the anatomical damage in glaucoma commonly precedes the functional one, and previous studies have demonstrated that up to 1/3 of the RGCs may be lost before the onset of VF defects [28], the early glaucoma diagnosis and treatment appear to be of fundamental importance.

Moreover, data acquired using magnetic nuclear resonance imaging support the hypothesis that glaucomatous damage is not limited to the eye, but also involves visual (lateral geniculate nucleus and visual cortex) and nonvisual areas of the central nervous system (CNS) [29]. The CNS alterations could be the result of the anterograde trans-synaptic transmission of the RGCs death signals or the consequence of a retrograde pathway induced by a neurodegenerative disease primarily affecting the CNS [29].

Although several possible mechanisms of RGCs death have been proposed and many different risk factors have been identified, the primum movens of GON remains unclear, so that POAG and NTG are seen as part of a continuum spectrum of open-angle glaucomas. Ocular hypertension (OHT) is considered the most important risk factor in the development and progression of POAG, whereas IOP-independent variables are believed to play an important role in the pathophysiology of NTG [1,30,31,32]. 

Several theories have been proposed to explain the NTG onset and progression, which can be divided into IOP-dependent and IOP-independent ones. 

### 2.1. The IOP-Dependent Theories 

Although multiple genetic and environmental risk factors for GON onset and progression have been identified, OHT is considered to be the most important one that includes NTG, which also happens to be the only modifiable factor [1,2,30,31,32]. It is thought that an elevated IOP may damage the RGCs axons at the site of the lamina cribrosa directly, by a mechanical effect, or indirectly, by reducing the blood supply. According to the mechanical theory, an elevated IOP induces stress on the lamina cribrosa, which is susceptible to compression and deformation, with subsequent damage to the RGCs axons and disruption of the axonal transport. The vascular theory, however, states that the OHT results in a reduced blood supply of the ONH, with ischemia and hypoxia of the RGCs axons [1,2,30,31,32].

Considering that the IOP is by definition, constantly within the normal range in NTG, two different theories have been elaborated to explain the occurrence of GON as a consequence of an IOP-related effect:


(1) The theory of the susceptibility to “normal” IOP values 


The “statistically normal IOP value” has been defined on the basis of large Western epidemiological studies to be about 16 mmHg ± 3 mmHg. The value of 22 mmHg has been historically used as the cut-off point to split normal and abnormal IOP values and to identify subjects requiring an ocular hypotensive treatment [1,2]. Several studies have demonstrated that different individuals show susceptibility to ONH damage at different IOP levels. Some subjects can develop OHN damage at “normal” IOP values, while others can tolerate IOPs higher than 21 mmHg [1,2]. According to this theory, it is believed that NTG patients may have an extremely higher individual susceptibility to IOP-related damage of RGC axons than normal eyes and could therefore develop ONH damage even in the presence of normal IOP values [30,31,32].

Supporting this theory, NTG patients have shown a larger circadian IOP fluctuation as compared to normal subjects [33]. Studies have demonstrated that, although within the statistically normal range, the IOP mean value, range and amplitude and frequency of nocturnal IOP spikes in the supine position are significant risk factors for glaucoma progression in NTG patients [34,35].


(2) The theory of the misclassification 


According to this theory, NTG patients may have undetected IOP spikes. As mentioned above, factors leading to an IOP underestimation, inducing, therefore, an overestimation of NTG patients, include as follows:Diurnal, nocturnal and postural IOP variations: in measuring the blood pressure and IOP every 3 h over 24 h, studies have [36] found that approximately 25% of NTG patients showed IOP spikes during non-office hours. Clinical trials based on the assessment of the nychtemeral IOP curves with a telemetric sensor incorporated in contact lenses (the Sensimed Triggerfish device) by Agnifili et al. [37] showed that 40–80% of the NTG patients had IOP spikes during the night.Corneal thickness variations: several studies have demonstrated that central corneal thickness (CCT) tends to be lower in NTG patients than in healthy subjects or in other types of glaucoma [38].Corneal biomechanics variations: in comparison with POAG and normal subjects, NTG patients have been demonstrated to have higher corneal deformability and lower corneal hysteresis (CH), i.e., the ability of the cornea to resist deformation [39].

### 2.2. The IOP-Independent Theories 

Considering that the development and progression of NTG theoretically occur in the absence of OHT, it sounds reasonable to suspect that IOP-independent mechanisms may play a fundamental role [17,23,24,25,30,31,32,36,40,41]. Several models have been proposed to explain the pathogenesis of GON in NTG patients, and each of them could become a potential target to promote the survival of the damaged RGCs. 

The most accredited theories are as follows: 


(1) ONH vascular perfusion insufficiency


One of the most important and popular pathophysiological theories suggests that GON in NTG patients could be related to vascular alterations, resulting in insufficient or fluctuating ONH and peri-papillary blood flow which induce an ischemia-reperfusion syndrome or transient hypoxia episodes [23,24,36,42,43]. 

The ONH is vascularized by branches of the ophthalmic artery, the terminal portion of the internal carotid artery, called short posterior ciliary (SPC) arteries. The SPC arteries arise from the ophthalmic artery as it crosses the optic nerve and divide into 10–20 branches. It is generally accepted that the prelaminar region of the ONH is supplied by the peripapillary choroid flow composed of the scleral short posterior ciliary system and the recurrent choroidal arteries; the lamina cribrosa, which is considered to be the precise site of the RGCs axons damage in glaucoma, is supplied by the centripetal branches of the short posterior ciliary arteries either directly or by forming the Zinn and Aller’s circle, and the retrolaminar region has a vascular supply from the pial vessels [1,2].

The most important concept related to ONH vascularization is that of the ocular perfusion pressure (OPP), defined as the difference between the mean systemic (both systolic and diastolic) arterial blood pressure and IOP [42]. The level of the OPP is considered directly related to the ONH blood flow [42]. By definition, both systemic hypertension and IOP reduction may result in an increased OPP, i.e., an enhanced ONH blood flow; conversely, systemic hypotension or elevated IOP values could induce an impairment of the ONH blood supply. These speculations can thus permit one to consider IOP-dependent and vascular IOP-independent theories as a continuum, with different weights in different patients. 

Supporting the so-called “ONH vascular perfusion insufficiency theory”, statistically significant relationships have been found between the prevalence of NTG and several ocular and systemic conditions that can induce or suggest a reduced OPP and an ONH ischemia or hypoxia, including: History of cardiovascular events or the presence of cardiovascular and/or cerebrovascular diseases, such as chronic atherosclerosis, obstructive arterial disease, intermittent claudication, vascular dementia, cerebral cortical micro-infarcts, atrial fibrillation, systemic hypertension or hypotension, in particular low diastolic blood pressure, excessive dip in nocturnal blood pressure levels and low mean OPP [36,42,43,44];Raynaud phenomenon and Flammer syndrome [45]. The latter is a recently described clinical entity characterized by a general dysregulation of the blood supply, with several different clinical manifestations, including cold hands and feet, low blood pressure and high blood pressure drops at night [45];Migraine [46];Obstructive sleep apnea syndrome (OSAS) [47];Impaired glucose tolerance, diabetes, smoking and high body mass [48];Ocular vascular alterations; reduced blood flow velocity in retrobulbar and peripapillary ciliary arteries; small central retinal vessel diameter; ophthalmic artery stenosis; malformations or branch occlusions; anomalous posterior ciliary arteries; high prevalence of circumpapillary atrophy, which is related to a deficit of the SPC arteries blood flow and disc hemorrhages, which seem to be related to platelet dysfunction [49];High aqueous and plasma levels of endothelin-1 (ET-1), a potent endogenous vasoconstrictor synthesized and released from the ciliary processes and involved in the regulation of the ocular blood flow [50].

All these data suggest that the vascular dysregulation and vasoconstriction mechanisms could be deeply involved in the pathogenesis of the NTG, so that GON in NTG could be caused mainly by reduced ocular blood flow instead of an elevated IOP [24,36,43]. It is important to note that a high percentage of NTG patients show systemic issues, including cardiac disease and OSAS, which may increase the morbidity and mortality in these patients [36]. 


(2) Apoptosis (self-killing) and autophagy (self-eating)


These are two forms of programmed cell death, which are essential in the maintenance of homeostasis and growth of the tissues. In particular, autophagy is a process by which eukaryotic cells regulate the turnover of long-lived proteins and cytoplasmic organelles. This process could have a protective effect in helping the neurons to eliminate damaged proteins and recycle the by-products for new synthesis. Both processes can be activated by oxidative stress and are thought to be up-regulated in NTG patients.

Growing evidence from animal model studies supports the involvement of the dysregulation in RGCs autophagy and apoptosis in the pathophysiology of NTG [24,26].


(3) Excitotoxicity


This is defined as a pathological process in which neurons are damaged by the hyper-activation of the receptors for the excitatory neurotransmitter glutamate, which can be induced by several factors, including hypoxia, ischemia and oxidative stress. The over-stimulation of the glutamate receptors allows high levels of calcium ions to enter the cell, inducing the activation of several enzymes that damage cell structures, such as the cytoskeleton, cell membrane and nucleic acids with consequent neuronal death, including the RGCs [24,51]. 

High levels of glutamate have been found in the RGCs of glaucoma animal models [24,40] and have been correlated to the RCGs’ death and glaucoma progression [22,26,27,51].


(4) Neurotrophins deficit 


Neurotrophins are proteins belonging to a class of growth factors involved in the differentiation, functioning and survival (inhibition of apoptosis) of the neurons. One of the most accepted theories to explain the RGCs loss with elevated IOP is that the OHT can cause an axonal transport obstruction of the ONH axons, which includes the transport of neurotrophins to the RGCs body: The lack of neurotrophins may start the RGCs apoptosis process [26,52]. 

Supporting this theory, serum and aqueous levels of brain-derived neurotrophins were found to be lower in NTG patients than in healthy subjects and POAG patients [53].


(5) Oxidative stress and the mitochondrial dysfunction


The reactive species or free radicals, such as the reactive oxygen species (ROS) and the nitrogen reactive species (RNS), are a class of substances mainly formed as a by-product of the oxidative phosphorylation of the mitochondria or through the activation of the glutamate receptors [41]. Several endogenous antioxidants are able to neutralize the damage caused by the free radicals through electron donation, including glutathione, superoxide dismutase and Coenzyme Q10. In the oxidative stress status, when the production of ROS and RNS is higher than the antioxidant species synthesis [41], free radicals can react with several cellular constituents, inducing DNA breakdown, proteins conversion, lipid peroxidation and mitochondrial functions inhibition, with consequent cellular aging and death [41]. 

Due to their high metabolic demands, RGCs are heavily dependent on high mitochondrial function [54], which is always associated with elevated ROS and RNS production as a consequence of the mitochondrial oxidative phosphorylation process [41]. 

Several studies have associated oxidative stress and RGCs death in glaucoma, especially in NTG [22,26,27,41,54]. The most important data supporting this theory are listed as follows:RGCs of NTG patient show decreased levels of endogenous antioxidants, such as glutathione [41].NTG patients are associated with specific mutations of the glutamate/aspartate transporter (GLAST) gene [41]. GLADT is expressed in the Mueller glia of the retina and removes excess glutamate from the synapses, preventing excitotoxic damage to the surrounding neurons. Loss of GLADT in mice leads to RGCs degeneration that simulates NTG [41].The risk of development of NTG has been associated with distinct features of mitochondrial DNA variations [54].As compared with normal matched-age subjects, the prevalence of NTG is significantly higher in patients affected by neurodegenerative diseases, such as Alzheimer’s or Parkinson’s disease, that are known to be associated with specific mitochondrial genetic variants [40,54,55].Optic neuropathies requiring a differential diagnosis from NTG, such as Leber’s hereditary optic neuropathy and autosomal dominant optic atrophy, are caused by mitochondrial genetic variants [54].


(6) Inflammation


It is suggested that low-grade chronic inflammation in response to various factors, including vascular dysfunction, ischemia, hypoxia, oxidative stress and elevated IOP, could participate in the pathogenesis of NTG. 

Several in vitro studies have demonstrated that the expression of many genes encoding pro-and para-inflammatory components, including metalloproteinase, cytokines, interleukins, prostaglandins and tumor necrosis factor, is up-regulated in RGCs of glaucomatous patients, especially in NTG patients [56,57]. The accumulation of these pro-inflammatory substances may induce RGC death [22,26,27,56,57]. 

Moreover, studies on glaucoma animal models and on ONH specimens of post-mortem glaucomatous eyes have demonstrated that during the early glaucomatous stages, prior to any detectable morphological damage, significant recruitment and activation of astrocytes, microglia, Mueller cells and factors of the complement system, which mainly regulate the retinal inflammatory response, can be detected at the RGCs layer and ONH sites [57].


(7) Autoimmune reaction


The theory of “autoimmune glaucoma” postulates the involvement of the immune system in the pathogenesis of glaucoma, with the production of autoantibodies against retinal antigens, such as rhodopsin and glutathione S-transferase, or against the proteoglycans of the ONH [58].

Supporting this hypothesis, several authors have found that 30% of NTG patients had one or more immune-related diseases compared to only 8% of controls [58]. Moreover, NTG patients have been shown to have an increased incidence of auto-antibodies against small heat shock proteins, Sjogren antigens, rhodopsin, glycosaminoglycans, etc. [59].


(8) Low intracranial and cerebrospinal fluid pressure


This theory assumes that the glaucomatous damage of the ONH axons can be caused by a higher pressure gradient between intracranial pressure and IOP at the level of the lamina cribrosa [30,60]. It is therefore supposed that a low cerebrospinal fluid pressure can increase the pressure gradient across the lamina cribrosa, inducing a GON even in the presence of normal IOP values. 

Although the reason is still unclear, NTG patients have exhibited cerebrospinal fluid pressure lower than that of normal subjects [60,61]; furthermore, previous studies have reported that cerebrospinal fluid diversion procedures are risk factors for developing NTG [61].


(9) Other theories


-The supine position: It has been demonstrated that moving from the sitting to the supine position induces an increase in IOP of approximately 4 mmHg in glaucomatous patients, with a further increase in IOP for lateral decubitus [1,2]. Previous authors have demonstrated that the VF indices in progressive NTG patients were related to the IOPs measured in the supine position, but not to those taken in the sitting position [62].-The abnormal eye movements: As known, eye movements can cause mechanical traction of the ONH. Using magnetic nuclear resonance imaging, previous authors have demonstrated that NTG patients may have abnormal globe retraction during eye movements, which could be related to the progression of glaucomatous damage [63].-The decreased scleral stiffness: It has been hypothesized that a lower scleral rigidity could be associated with higher deformation of the lamina cribrosa and subsequently greater axonal damage in response to the IOP. Unfortunately, scleral stiffness is difficult to measure in vivo. It has been suggested that the biomechanical properties of the sclera might correlate to those of the cornea, which are measurable in vivo with relatively new instruments, such as the Ocular Response Analyzer (ORA) and the Corvis-ST pachy-tonometer [20]. Considering that, as mentioned above, NTG patients have shown lower CCT values, higher corneal deformability and lower CH in comparison with healthy and POAG subjects [38,39], it is speculated that they may have also less resistant lamina cribrosa and peripapillary sclera tissues, which could potentially reduce the ability of these structures to dampen IOP changes. This might increase the susceptibility of the ONH to increased IOP or IOP spikes, with the consequent development of GON [64]. Supporting this theory, thin CCT and low CH have been indeed identified as risk factors for the development and progression of glaucoma [64].

## 3. The Therapeutic Approaches in NTG

### 3.1. Studies Supporting the IOP-Dependent Therapies: The Role of IOP Reduction in the Treatment of NTG

Filtering surgery studies during the 1980s and 1990s have shown that a substantial lowering of IOP can reduce the progression of damage in NTG patients [65]. In more recent years, large clinical trials have demonstrated that reducing IOP is effective in delaying the progression and probably preventing the onset of VF damage in NTG patients [66,67], strongly supporting the central role of the IOP reduction in the NTG therapy and the rationale for further pressure reduction in those cases showing a damage progression despite low IOP values [30]. 

The Collaborative Normal-Tension Glaucoma Study (CNTGS), a multicenter, prospective, randomized, double-masked and controlled clinical trial including 140 NTG patients with documented VF damage progression, demonstrated that, after a 7-year follow-up, an IOP reduction of at least 30% resulted in a stabilization of the VF defects in 88% of cases as compared to the 65% of untreated patients [66].

The Early Manifest Glaucoma Trial (EMGT), a randomized double-masked clinical trial conducted on 255 OAG patients with early VF defects, including 53% of NTG cases, showed that a 25% reduction in IOP can reduce the risk of disease progression to 45% in the study group, compared to 62% in the control group, after a 6-year follow-up [67]. 

These studies have suggested that, to be significantly effective in decreasing the risk of NTG progression, the reduction in IOP should be at least 30% or more [65,66]. 

Moreover, studies with long follow-ups have shown that the amount of IOP reduction seems to be directly related to the reduction in the VF progression rate in NTG patients [68]. Additionally, the benefits of the IOP reduction in NTG patients seem to be significantly higher in females with migraine and a family history of glaucoma, without disk hemorrhages, family history of stroke and personal history of cardiovascular disease [69].

### 3.2. Studies Supporting the IOP-Independent Therapies for the Treatment of NTG

The Collaborative Normal-Tension Glaucoma Study (CNTGS), the most important study supporting the usefulness of decreasing the IOP in NTG patients, showed that 12% of NTG patients showed disease progression despite a 30% IOP reduction from baseline [66]. The Early Manifest Glaucoma Trial (EMGT) showed that despite a 25% reduction in IOP from baseline, 55% of patients experienced progression of VF damage [67].

Both the CNTGS and EMGT, although confirming the importance of reduction in IOP in patients with NTG, clearly simultaneously showed that the IOP reduction alone is not enough to slow or halt the progression of glaucoma damage in certain subjects [66,67], suggesting that other IOP-independent risk factors may contribute to the pathogenesis of NTG [23,30,31,32,36] and that the shift toward the IOP-independent therapies may have a rationale [23,24,25].

Moreover, the Low-Pressure Glaucoma Treatment Study (LoGTS), a prospective, randomized clinical trial in which NTG patients were randomized to receive timolol or brimonidine eye drops as monotherapy, showed that despite similar IOP values in both groups, the VF loss was significantly less frequent in the brimonidine-treated group [70], strongly suggesting that an effective anti-glaucomatous treatment should not only reduce IOP but also offer IOP-independent therapeutic pathways, such as neuroprotection and regeneration of the RGCs.

## 4. The Pharmacological Approaches in NTG Therapy

### 4.1. IOP-Dependent Therapies: The Drugs Used to Reduce the IOP in NTG

The literature data strongly indicate that the mainstay of current NTG treatment is the reduction of IOP [30,68], and the first-line treatment involves the use of hypotensive eye drops [68]. IOP can be lowered medically either by decreasing the aqueous production or by increasing the trabecular or the uveo-scleral outflow pathways.

The drugs currently used in NTG therapy for their pre-eminently IOP-lowering effect are listed in Table 1.

-**Prostaglandin (PGAs)F2α analogs and prostamide analogs** (latanoprost, travoprost, bimatoprost and tafluprost): Since the introduction of latanoprost in 1996, these drugs have become the first-line treatment in POAG and NTG medical therapy because they have shown the highest efficacy in reducing IOP with adequate diurnal and nocturnal IOP control, good safety profile and the convenience of only one application per day, which results in higher patient’s compliance [68,71,72,73]. 

These medications induce an IOP reduction of approximately 25–33% by increasing the uveo-scleral outflow secondary to the widening of the ciliary muscle and, to a lesser extent, by enhancing the trabecular meshwork outflow [72,73]. 

These medications demonstrated to be able to lower the IOP over 24 h and to reduce the nocturnal IOP peaks [72], which are considered to be one of the risk factors in the progression of NTG [35]. In particular, latanoprost has been shown to maintain the IOP reduction and the enhancement of the uveo-scleral outflow during the nighttime, whereas the ocular hypotensive effect of the timolol is abolished during the nocturnal/sleep period [71,74]. 

PGAs and prostamide analogs have shown good long-term safety profiles. Potential adverse effects include increased eyelash growth, periocular hyperpigmentation, palpebral and conjunctival hyperemia, allergic conjunctivitis, keratitis and herpes virus activation; most importantly, the systemic side effects are rare and minimal. Nonetheless, because of their pro-inflammatory effect, the use of these drugs is discouraged in patients with active ocular inflammation and cystoid macular edema [73].

Moreover, PGAs analogs were demonstrated to increase the ocular blood flow in NTG patients [75]. Latanoprost, bimatoprost and tafluprost have demonstrated a neuroprotective effect on the RGCs in vitro and in animal models [76].

-**Nitric-oxide-donating PGAs analogs:** Latanoprostene bunod is a nitric-oxide-donating prostaglandin F2α analog that has recently received approval from the Food and Drug Administration (FDA) in the US for reducing IOP in patients with OAG or OHT, which will be available in Europe in the near future [77]. 

This drug increases the aqueous outflow both by uveo-scleral and trabecular pathways [77]. 

Multicenter, randomized, double-masked, phase III clinical trials have demonstrated that the latanoprostene bunod is effective in lowering the IOP in OAG and OHT patients both during the diurnal/wake and nocturnal/sleep periods, showing a lowering effect significantly higher than that of latanoprost and timolol, with a side effect profile similar to that of prostaglandin analogs [77].

-**Beta-adrenergic antagonists (beta-blockers)** (timolol, levobunolol, carteolol, betaxolol, carvedilol and nebivolol): Since the introduction of timolol for glaucoma therapy in 1979, these drugs have been considered as the first-line ocular hypotensive therapy for approximately 25 years [78]. 

Beta-blockers can be divided into nonselective beta-adrenoceptor antagonists, including timolol, carteolol, levobunolol and carvedilol, and selective beta-1-adrenoceptor antagonists, including betaxolol and nebivolol, which have been demonstrated to induce fewer side effects on cardiac and pulmonary functions in comparison with nonselective beta-blockers [78,79].

Beta-blockers induce a diurnal IOP reduction of 20–25% by decreasing aqueous humor production from the epithelial cells of the ciliary body [78]. Sleep laboratory studies have demonstrated that beta-blockers do not have any IOP-lowering effect during the nocturnal/sleep time [42,71,74,78]. Due to a decrease in endogenous circulating catecholamine levels, the aqueous humor production is significantly lower during the night, which may explain the decreased nocturnal efficacy of the beta-blockers [42,71,74,78].

Potential side effects of the beta-blockers include allergic conjunctivitis, keratitis, bronchospasm, vasospasm, bradycardia, systemic systolic and diastolic hypotension and heart rate reduction [78,79].

Carteolol is a nonselective beta-adrenoceptor blocker with partial agonist activity [80]. These eyedrops, in comparison with those of timolol, although showing comparable IOP-lowering efficacy, lack local anesthetic activity, with consequently less ocular surface irritation, and induce a reduced decline in heart rate or dyspnea, likely due to the partial agonist activity of the carteolol [80].

Levobunolol is a nonselective beta-blocker that has shown an IOP-lowering efficacy similar to that of timolol and a concomitant vasodilatory effect on the vascular smooth muscle cells, likely due to the block of calcium channels [81].

Betaxolol is a selective beta-1-adrenoceptor antagonist [82]. The use of this eyedrop in NTG is controversial. Studies have shown that similar to timolol, betaxolol can induce hypotensive dipping of blood pressure during nighttime, with detrimental effects on the VF damage progression [83]. Comparative clinical trials investigating the use of local beta-blocker for managing NTG have shown a greater reduction in IOP with timolol, however, better VF preservation using betaxolol [82], suggesting that the decrease in blood pressure due to the systemic effect of nonselective beta-blockers could have a negative effect on the preservation of visual function. 

Carvedilol is a relatively new nonselective beta-blocker with multiple other actions, including antioxidant activity, vasodilatation, inhibition of apoptosis, anti-inflammatory activity, calcium channel blocking and mitochondrial protective effects [84]. Studies have shown that this drug reduces IOP following both topical and oral administration in vivo in animal models [85,86].

Nebivolol is a novel beta-1-selective adrenoceptor antagonist that has been shown to decrease IOP following both topical and oral administration in animal models [85].

Betaxolol, carteolol and levobunolol eye drops have been associated with increased ONH blood flow in glaucoma patients [78,80,81,82]. Carvedilol has shown the ability to improve ocular microcirculation in animal models after topical or oral administration by blocking the alfa-adrenergic receptors [85]. Nebivolol has been shown to improve the ocular blood flow in rabbits [85] and in glaucoma patients suffering from concomitant arterial hypertension after oral administration [87]. The effect of nebivolol on ocular hemodynamics is likely related to its known peripheral vasodilatory effects due to its NO-releasing properties [87].

Beta-blockers have shown neuroprotective properties in vitro and in animal models, which have been proposed to be related to their ability to reduce the amount of glutamate entering and damaging the RGCs and to their calcium-channel-blocking properties [25,78,84,86].

In general, the use of beta-blockers in NTG is debatable because of their possible negative effect on the ONH blood flow. These drugs have known vasoconstrictive properties. Moreover, the absorption of topical beta-blockers into the systemic circulation and the administration of an evening oral dose of beta-blockers have been demonstrated to increase the physiologic nocturnal arterial systolic and diastolic hypotension and to reduce the heart rate and the blood oxygen saturation [42,71,78]. Supporting these concerns, previous studies have demonstrated that the treatment of NTG patients with timolol or betaxolol increases the risk for VF deterioration [83]. 

Recent clinical trials on NTG patients have shown that betaxolol and carteolol had a protective effect on the VF indices which was IOP-independent [88].

-**Selective alpha-2-adrenergic-agonist** (Brimonidine): Brimonidine has shown the ability to reduce the IOP production and to improve the uveo-scleral outflow consequent to a slight pupil dilatation [78,89]. 

The IOP-lowering effect induced by brimonidine is approximately 20–25% [89], and it appears to be effective only during the diurnal/wake period, whereas it appears minimal during the nocturnal/sleep period [89]. 

Because of its high selectivity for alfa-2 than for alpha-1 adrenergic receptors, the brimonidine did not induce mydriasis and vasoconstriction of the retinal vessels [78,89]. Possible side effects are allergic and follicular conjunctivitis, dry mouth and nose and systemic hypotension [89]. Quaranta et al. found a greater reduction in mean 24 h systolic and diastolic blood pressure with brimonidine than with timolol [42]. 

Brimonidine has shown the ability to inhibit the apoptosis of the RCGs in vitro and in vivo animal models possibly through the up-regulation of the neurotrophins, in particular, the so-called brain-derived neurotrophic factor (BDNF) [25,40].

Moreover, the Low-Pressure Glaucoma Treatment Study (LoGTS), a multicenter, prospective, randomized, double-masked clinical trial in which 190 NTG patients were randomized to receive timolol or brimonidine eyedrops as monotherapy, showed that patients treated with brimonidine were less likely to show VF loss progression than those receiving timolol, despite similar IOP reductions in both groups [70], suggesting that brimonidine could provide an adjunctive neuroprotective effect beyond IOP-lowering or, alternatively, that timolol could have a neuro-destructive action, likely related to its lowering effect on systemic blood pressure and pulse rate [70,78]. It is important to note that approximately one-third of the patients receiving brimonidine in the LoGTS stopped the therapy because of side effects, especially allergic conjunctivitis [70].

Furthermore, a recent clinical trial on NTG patients showed that brimonidine, and also betaxolol and carteolol, had a protective effect on the VF indices which was independent of the IOP-lowering efficacy [88].

-**Carbonic anhydrase inhibitors** (CAIs) (dorzolamide, brinzolamide and acetazolamide): They inhibit the carbonic anhydrase isoenzyme 2, reducing the IOP production, with a baseline IOP reduction of approximately 15–20% for topical and 20–30% for oral administration [90,91]. The CAIs have been demonstrated to lower IOP for 24 h, also during the nocturnal/sleep period [74,90,91]. 

Potential adverse effects include allergic dermatitis, corneal edema, Stevens–Johnson syndrome, malaise, anorexia, depression and renal calculi [90]. 

Topical dorzolamide has been shown to induce a significant improvement in almost all hemodynamic parameters of intraocular and periocular vessels in both normal and glaucomatous eyes [90,91]. Moreover, dorzolamide has demonstrated a neuroprotective effect in reducing the RGCs apoptosis in vitro [91].

-**Miotics** (Pilocarpine): This agent, an alkaloid isolated from a South American plant (Pilocarpus jaborandi), is the first known topical anti-glaucoma medication introduced in 1875. Pilocarpine lowers the IOP by traction of the scleral spur induced by an increased parasympathetic tone of the ciliary muscle, resulting in enhanced trabecular outflow [71,92]. The mean IOP reduction is approximately 20–25% [71,92]. 

Possible side effects are increased myopia, decreased vision, cataracts, periocular contact dermatitis and ocular congestion [71,92]. Moreover, treatment with pilocarpine requires administration four times daily, resulting in low patient compliance. Pilocarpine is deemed to have a neuroprotective effect by maintaining calcium homeostasis and mitochondrial membrane integrity [25].

Previous authors, investigating the IOP-lowering effect of pilocarpine as monotherapy in NTG patients, have shown modest efficacy, with an IOP reduction of at least 30% from baseline in only 13% to 27% of cases [93]. 

-**Rho-associated protein kinase (ROCK) inhibitors**: The ROCK inhibitors, many of which are actually under phase II and phase III studies, reduce IOP by disrupting the cytoskeleton and relaxing the smooth-muscle-like cells of the trabecular meshwork and Schlemm’s canal tissues, thus increasing the aqueous humor trabecular outflow [77,94,95]. The mean IOP reduction induced by the ROCK inhibitors is approximately 10–20% [94,95].

ROCK inhibitors have shown a high incidence of local adverse effects, including conjunctival hyperemia, eye pain, irritation, pruritus and discharge, whereas no systemic side effects are reported [77,94,95]. 

The ROCK inhibitors have also been shown to increase the ONH blood flow via vasodilatation of the ciliary arteries [94]. Moreover, they have demonstrated a neuroprotective effect on the RGCs and a regenerative effect on the ONH axons through the modulation of the RGC apoptosis in both in vitro and animal models [25].

A prospective, multicenter, randomized, double-masked, placebo-controlled phase II study evaluating the hypotensive efficacy and safety of the sovesudil, a novel ROCK inhibitor, in NTG patients, showed that a 0.5% ophthalmic solution of sovesudil administered three times daily has a statistically significant IOP-lowering effect as compared with placebo, with mild adverse events, including conjunctival hyperemia in 25% of cases [95]. 

Netarsudil is a potent Rho kinase/norepinephrine transporter inhibitor that has recently received approval from both the FDA and the European Medicines Agency (EMA) for treating patients with OAG and OHT, and it will be shortly available on the market [77]. 

It acts by decreasing the aqueous production, increasing the trabecular outflow and possibly decreasing the episcleral venous pressure [96]; moreover, it has the convenience of a once-daily dosage.

Multicenter, double-masked phase III clinical trials have demonstrated the non-inferiority of netarsudil when compared to timolol in the treatment of OAG and OHT patients [96]. Moreover, netarsudil has shown to be more effective in subjects with baseline IOP ≤ 26 mmHg, likely because of its ability to decrease the episcleral venous pressure [96], suggesting a rationale role in the NTG therapy. 

The reported local side effects of netarsudil include conjunctival hyperemia (50–60% of cases), subconjunctival hemorrhages, cornea verticillata, eyelid erythema, increased lacrimation, instillation-site pain and blurred vision, whereas no systemic adverse effects were observed [96]. 

-**Combination therapy**: The use of more than one drug is needed in NTG patients in which a 30% reduction in IOP is not achieved with monotherapy, leading to an increased frequency of eye-drop administrations and possible side effects and higher costs. Fixed drug combinations have the advantage of combining substances with additive mechanisms of action and reducing the number of drops administered, showing therefore higher efficacy and better compliance than mono- or multiple drugs.

Fixed combinations of dorzolamide/timolol and brimonidine/timolol have been demonstrated to be safe and effective IOP-lowering drugs in NTG patients, without affecting the ocular perfusion [71].

The fixed combination of netarsudil and latanoprost has been recently approved by the FDA for the treatment of OAG and OHT patients with only one daily administration [77]. This drug combination is the only glaucoma medication that acts on all IOP-reduction mechanisms, i.e., increasing both trabecular and uveo-scleral outflow and decreasing aqueous production and episcleral venous pressure. Recent multicenter, randomized, double-masked clinical trials on OAG and OHT patients have demonstrated that the fixed combination of netarsudil and latanoprost provides an IOP-lowering effect of ≥30% and local side effects similar to its individual components [77].

### 4.2. The IOP-Independent Therapies

The IOP-independent strategy for NTG therapy is a relatively new concept that includes two main purposes: (1)Maintenance and/or increase of the ONH blood perfusion and/or oxygenation;(2)Neuro-protection, i.e., prevention and/or reduction of the degeneration and death of the RGCs, and/or RGCs regeneration.

Substances targeting the axonal damage and the RGCs apoptotic cascade, having therefore antioxidant, anti-inflammatory, anti-apoptosis or vaso-active properties, may therefore represent an alternative approach to the NTG treatment [23,24,25,36,40,41,97,98,99,100,101,102].

The following section summarizes the drugs and dietary supplements that have shown the ability to increase OHN perfusion or to have neuroprotective properties on RGCs in vitro, in glaucoma animal models in vivo, or in preclinical or clinical trials on glaucomatous patients. 

#### 4.2.1. Dietary Supplements in the NTG Treatment

The “dietary supplements” are defined by the FDA as substances, including vitamins, minerals, botanicals, herbs or dietary elements, used to supplement the diet by increasing the total dietary intake, in order to correct nutritional deficiencies, maintain an adequate intake of certain nutrients or to support specific physiological functions [98,99,100,102]. In contrast to drugs, which are designed to treat illnesses or diseases, dietary supplements cannot provide any pharmacological, immunological or metabolic actions [89,90,91,92,93,94,95,96,97,98,99,100,102]. On the basis of this regulatory definition, the producers cannot therefore claim any clinical effect of dietary supplements on glaucoma unless these substances have been investigated and registered for use as a drug. Some of these substances are currently in phase II or III of drug development.

The dietary supplements that could have a rationale in the NTG treatment are summarized in Table 2 and are as follows:

-**Ginkgo biloba extract (GBE)**: It is a natural chemical compound found in the leaves of a tree indigenous to Korea, Japan and China [98,99,100,102,103]. Originally used in Chinese traditional medicine as a treatment for different medical conditions, the GBE was first introduced in Europe in 1965 known as EGb761. The commercially available GBE provides 60 bioactive compounds, mainly consisting of flavonoids (24% to 27% of the extract) and terpenoids (5% to 7% of the extract) [103]. 

The GBE has been demonstrated to have several different effects [98,99,100,102,103], including: Vasodilatation and enhancement of the cerebral blood flow;Antioxidant properties related to its free radical scavenging activity;Anti-inflammatory effect, inducing a decrease in the levels of the pro-inflammatory prostaglandins and cytokines;Regulation of mitochondrial activity, especially by reducing the mitochondria’s oxidative stress;Anti-apoptotic activity by the downregulation of pro-apoptotic genes;Hemorheological regulation effect, by increasing the erythrocyte deformability and because of a fibrinolytic effect;Neuroprotective activity: GBE provides neuroprotection against ROS, calcium overload, nitric oxide and beta-amyloid-induced toxicity and ischemic-reperfusion-inducing toxicity;Neurotransmission regulation, by regulating the gene expression of neurotransmitter receptors;Hormonal regulation: GBE increases the expression of several hormones, such as thyroid, growth hormones and prolactin, which are essential for neuronal proliferation and differentiation, cognitive capacity related to memory, alertness, motivation and working capacity, and it has been proven to be beneficial in the treatment of cognitive disorders, including dementia;Anti-neoplastic activity, by regulating the expression of proteins involved in DNA damage signaling, repair and gene expression.

Within the specific prescribed dosage, the GBE has shown minimal side effects, which include stomach upset, headache, dizziness, constipation, palpitation and allergic skin reactions [102,103].

GBE has shown neuroprotective capacity in experimental animal models of chronic glaucoma [102,103]. Several studies have investigated the effects of the GBE on NTG patients [98,99,100,102,103], showing the following data:Absence of statistically significant effect on the IOP values;Significantly higher peri-papillary blood flow when compared to placebo;Controversial results on the VF indices;Significant delay in the VF loss progression.

Despite promising results, the lack of rigorous registration studies does not allow to draw firm conclusions on the use of GBE in NTG [103].

-**Resveratrol**: It is a polyphenol commonly found in fruits such as berries and nuts and in the skin of red grapes and red wine, having many properties, including cardioprotective, neuroprotective, antioxidative, anti-inflammatory, antidiabetic and anti-tumoral effects [98,100,102,104]. Also, it has shown a neuroprotective effect on RGCs and their axons in cell culture and animal models [104]. 

-**Citicoline**: It is an endogenous compound crucial for cellular functioning, which undergoes rapid metabolism to form cytidine and choline [105,106]. It acts as an intermediate in the biosynthesis of cell membrane phospholipids and as a precursor for the neurotransmitter acetylcholine [106]. Furthermore, it increases the levels of acetylcholine, dopamine, noradrenaline and serotonin in several brain regions and the dopamine release in the retina [105]. 

Citicoline has a neuroprotective effect, due to the protection of the cell membranes, in particular the mitochondrial ones [105].

As a dietary supplement, citicoline has been used in many neurodegenerative diseases, including Parkinson’s and Alzheimer’s diseases, dementia, stroke and glaucoma [105,106], showing negligible toxicity [106].

Citicoline has demonstrated neuroprotective properties on the RGCs both in vitro and in glaucoma animal models [106].

Although previous studies failed to show any evidence of citicoline deficiency in glaucomatous patients [106], the intramuscular or oral administration of citicoline for at least 2 months in OAG patients has been associated with an improvement in all pattern electroretinogram (PERG) and pattern visual evoked potential (VEP) indices [98], whereas the treatment with citicoline eyedrops was associated with a significant slowing down of the VF damage progression [107]. 

Based on these results, citicoline has been approved by the European Union and the Italian Ministry of Healthy as a dietary food supplement for special medical purposes in glaucoma patients [106].

-**Coenzyme Q10 or Ubiquinone-10**: It is an important cofactor of the mitochondrial electron transport chain and a potent antioxidant [41,98,99,102,108]. It is predominantly present in animal organs, especially in the heart, liver and kidney, and it is also found in several foods, like meat, fish, soy oil and peanuts [98,102,108]. Due to the large molecular weight and lipid-solubility, ubiquinone-10 has poor intraocular penetration, and for these reasons, it is usually used in combination with vitamin E, which increases its bioavailability [98,102,108].

Several in vivo and in vitro studies have demonstrated its anti-apoptotic effect on the RGCs in glaucoma models [25,40,108]. 

OAG patients treated with a topical combination of coenzyme Q10 and vitamin E for at least 6 months have shown a significant enhancement of both PERG and VEP responses [109]. 

-**Camellia sinensis (L.) Kuntze or Green tea**: Green tea beverage is made from the infusion of the leaves of Camellia sinensis and contains flavonoids, especially catechins, and alkaloids (such as caffeine and theobromine). The major catechin present in green tea is epigallocatechin-3-gallate (EGCG), which is also found in high concentrations in red wine, dark chocolate, legumes and nuts.

Both EGCG and the green tea extract in toto have shown anti-apoptotic, antioxidant and anti-inflammatory properties on the RGCs in vivo and in vitro models [98,99,100,102]. 

A clinical study showed that the oral administration of EGCG induced an increase in the PERG indices in OAG patients [110].

-**Panax Ginseng**: It is one of the most used medical herbs in Asia [98,100,102]. 

It has demonstrated antioxidative and anti-apoptotic properties on the RGCs in vitro and in animal models [98,100,102].

The diet supplementation of OAG patients with Panax Ginseng showed significant retinal peripapillary blood flow improvement [98,100,102].

-**Anthocyanins**: They are a kind of polyphenols abundant in berries, currants, grapes and some tropical fruits, which have been demonstrated to have antioxidant, anti-inflammatory, anti-cancer, anti-angiogenesis, anti-diabetic, anti-obesity and anti-microbial effects [100,102,111]. 

They have been shown to increase the survival of RGCs in both in vitro and in vivo models [102] and to be able to normalize the serum endothelin-1 levels in glaucoma patients [100]. 

A pilot study found that the administration of black currant anthocyanin in NTG patients for 6 months induced a significant increase in ONH and peripapillary retina blood flow, with no significant IOP and VF changes, and that the intake for 2 years reduces the VF deterioration in NTG patients compared to placebo, with no IOP changes [111]. 

-**Cannabinoids**: They are compounds found in the Cannabis sativa plant, commonly known as marijuana, and represent one of the most used psychoactive substances in the world. Cannabinoid-like substances released by the neurons and referred to as endocannabinoids have been discovered in 1992 [102,112]. 

The cannabinoids exert their effects by interacting with specific endocannabinoid receptors present in the central nervous system [112] and, depending on the brain area involved, include alteration of memory, cognition, psychomotor performances, pleasure responses and pain perception.

In both in vitro and in vivo studies, the cannabinoids have been demonstrated to decrease IOP by both reducing the production and increasing the aqueous humor outflow and also protecting the RGCs against glutamate-induced excitotoxicity [40,113]. 

Furthermore, a significant transient IOP reduction from 30 min to 4 h after the cannabinoid inhalation has been demonstrated in both OAG patients and healthy subjects [40,102,112,113].

-**Palmitoylethanolamide (PEA)**: It is a lipid mediator synthesized during inflammation and tissue damage, with neuroprotective, anti-inflammatory and analgesic properties [98,114]. It is present in various foods (eggs, soybeans, peanuts, etc.), and it is marketed as a medical food at the dosage of 600/12,000 mg/day in several European countries, including Italy [114]. 

Based on its pharmacological properties, it is speculated that PEA can act by increasing the aqueous humor trabecular outflow, inducing vaso-relaxation in the ophthalmic arteries and stimulating the cannabinoid system [115]. 

PEA has demonstrated neuroprotective efficacy on the RGCs in animal models [114]. Previous authors have demonstrated that the systemic administration of PEA for at least 6 months in NTG patients reduced IOP and improved the PEV and PERG indices, without side effects [115]. 

-**Vitamins supplementation**: Vitamins are organic compounds and essential micronutrients found in plants and animals. The potential neuroprotective effect of vitamins is thought to be mainly related to their antioxidant activity [25,41,98,99,100,102]. 

Several previous studies have shown that, in comparison with healthy subjects, NTG patients have significantly lower levels of vitamin E, vitamin B3 (nicotinamide or niacin), retinol (vitamin A) and vitamin C [41,98,99,100].

The vitamins B3, B6, B12, C, D and E have shown a neuroprotective effect on the RGCs in vitro and in glaucoma animal models [24,41,98,99,100,102]. Previous authors have demonstrated that diet supplementation with 300 mg/day of vitamin E induced a statistically significant improvement in VF indices and blood flow in the ophthalmic and posterior ciliary arteries. On the other hand, the oral B12 supplementation in NTG patients did not show any significant reduction in VF damage progression during a 4-year follow-up [25,98,102].

-**Hydrogen sulfide (H2S)**: It is a gas-transmitter with several endogenous functions in mammalian tissues. It has been demonstrated to reduce the IOP by increasing the humor aqueous outflow, to scavenger ROS species and to increase the glutathione levels, protecting the RGCs from excitotoxicity [116]. Despite its potential, its use is limited by the delivery challenges related to its unique physiochemical properties [116].

-**Other substances**: Several other compounds have shown a neuroprotective effect on RGCs and their axons in cell culture or in animal models, in addition to beneficial biological properties in glaucomatous patients, which include lutein and zeaxanthin; nitric oxide; flavonoids; Crocus sativus L. or saffron; hesperidin; nicotine; ethylic alcohol; crocin and crocetin; zinc; magnesium; curcumin, spermidine; creatine; alfa-lipoic acid; apolipoprotein-E; nuclear factor-kappa B; omega-3 polyunsaturated fatty acids; melatonin; taurine; forskolin; Lycium barbarum; Erigeron breviscapus Hand. Mazz.; Scutellaria baicalensis Georgi; Diospyros kaki L.; Tripterygium wilfordii Hook F; caffeine [23,25,41,98,99,100,102].

-**Caloric restriction**: It has been shown to promote the RGCs cell survival in a mouse model of NTG likely by increasing the synthesis of neurotrophic factors, catalase and anti-apoptotic proteins and by decreasing the oxidative stress levels [117].

#### 4.2.2. Drugs Used for Their IOP-Independent Effects in NTG Patients

Several drugs, approved for the treatment of other diseases, have shown to be useful in the NTG treatment, being thus suitable candidates for drug repositioning. These substances, listed in Table 3, include the following:

-**Calcium channel blockers (CCBs)** (nifedipine, nimodipine and verapamil): These drugs were introduced in the 1960s for the treatment of systemic hypertension and other cardiovascular diseases [118]. Although the CCBs are the only FDA-approved treatment addressing the vascular risk factors in glaucomatous patients, their efficacy for NTG therapy is still debated [25,71,118]. They have several ocular effects [71,118], including the following:To improve the ONH and choroidal in healthy and glaucoma subjects, especially in NTG patients, by inducing vasodilatation in the posterior ciliary arteries, which has been demonstrated using color Doppler imaging and laser Doppler flowmetry;To slow the progression of VFs defects and ONH damage in NTG, whereas the efficacy in POAG patients is debated;To increase VF indices and color contrast sensitivity in NTG patients;To reduce the glutaminergic neurotoxicity on the RGCs both in vitro and in glaucoma animal models, suggesting neuroprotective properties;To induce systemic hypotension that may theoretically decrease the ONH perfusion. For this reason, the use of CCBs in glaucoma, especially in NTG, is controversial. Many authors suggest avoiding systemic anti-hypertensive medication and local beta-blockers at nighttime, because both beta-blockers and CCBs may have a negative impact on the perfusion and oxygenation of the ocular tissues [42,71,78,118].

-**Memantine**: This agent is a glutamate antagonist approved in Europe and the USA for the treatment of Alzheimer’s disease [24]. Although it has been demonstrated to slow the glaucomatous ONH damage progression in macaque monkeys [25], large prospective multicenter clinical trials in humans failed to show benefit in glaucomatous patients [119].

-**Angiotensin-converting enzyme inhibitors (ACEIs)**: These drugs, widely used for the treatment of systemic hypertension, have shown a significant IOP-lowering effect in both POAG and OHT patients and neuroprotective properties in vitro [23].

-**Anticonvulsants**: Drugs like valproic acid have been approved for clinical use in the treatment of various conditions, such as epilepsy, migraines and neuropathic pain. It has been tested in an experimental mouse model of NTG, showing the ability to protect RGCs from oxidative stress [41,120]. 

-**Edaravone**: It is a drug, acting as a free radical scavenger, used for the treatment of acute brain infarction and amyotrophic lateral sclerosis. It has been demonstrated to be effective in preventing the death of RGCs in an NTG mouse model [121].

-**N-acetylcysteine**: It is a drug historically used as an antidote against paracetamol overdose and more recently used for several medical conditions. It has demonstrated neuroprotective efficacy by its antioxidant properties in NTG animal models [41].

-**Statins**: These are a class of drugs used as anti-cholesterol medication. A prospective clinical trial showed that simvastatin provided a protective effect in NTG patients [23].

-**Androgen-deprivation therapy**: This therapy, used in patients with prostate cancer, has been associated with a lower incidence of newly diagnosed NTG, suggesting a role of testosterone in the pathogenesis of NTG [122].

-**Minocycline**: It is a second-generation tetracycline with well-recognized anti-inflammatory properties. It provided neuroprotection in glaucoma animal models by preventing the microglia activation and blocking the apoptotic cascade of the RGCs [25,57].

-**Azithromycin**: It is a macrolide antibiotic with anti-inflammatory activity. It has been shown to reduce the apoptosis of the RGCs in an in vitro experimental model of glaucoma [25,57].

-**cAMP phosphodiesterase inhibitors**: Ibudilast has been proposed as a potential therapy for several neurodegenerative diseases and is under investigation for the treatment of multiple sclerosis. It has been demonstrated to promote the survival of RGCs in glaucoma animal models [25,57].

-**Continuous positive airway pressure (C-PAP)**: Previous studies have demonstrated that the C-PAP therapy, in patients with both NTG and OSAS, is effective in stabilizing the VF defects, likely by improving ONH oxygenation and perfusion [123]. 

## 5. NTG Prognosis, Considerations about the Available Pharmacological Approaches, Future Perspectives and Conclusions 

The prognosis of NTG varies and depends on factors that include disease severity at diagnosis, effectiveness of treatment, individual risk factors, overall ocular and general health, etc. [5,35,69,124,125]. 

The results of the CNTGS provide useful information about the natural history of the NTG disease. This study showed that approximately 65% of untreated NTG eyes did not show any VF damage progression over 7 years of follow-up, suggesting that numerous NTG patients can be monitored without treatment, at least initially, or that, alternatively, have been misdiagnosed. Thirty-five percent of untreated NTG patients showed disease deterioration with a highly variable rate of VF loss progression [66]. 

When a medical, laser or surgical therapy is applied to NTG patients in order to reduce IOP of at least 25–30% from baseline values, disease stabilization was observed in 88% of patients in the CNTGS [66] and in 55% of cases in the EMGT [67]. 

The main risk factors associated with the NTG disease progression in both treated and untreated patients have been demonstrated to be female gender, greater variation in diurnal IOP and diastolic blood pressure, presence of disk hemorrhage, greater vertical cup/disc ratio and migraine at baseline. Age, mean IOP and baseline IOP were not shown to be risk factors for progression. Epidemiology studies have shown that Asians tend to show a slower rate of disease progression [35,69,124,125]. On average, the visual field damage progression has been reported to be slower in NTG than in POAG, but with higher inter-patient variability [5,124].

Similar to other types of glaucoma, NTG can progress to irreversible unilateral or bilateral blindness in the worst cases, even despite therapy [126]. The cumulative risk to develop legal unilateral blindness in treated NTG patients under standard ophthalmic care has been calculated to be 5.8% and 9.9% at 10 years and 20 years, respectively. The risk for bilateral blindness at 10 and 20 years was 0.3% and 1.4%, respectively [126]. Patients presenting advanced damage at the diagnosis or rapidly progressing VF loss, the so-called “rapid progressors”, are most likely to become blind due to the disease. It is fundamental that these patients need to be identified and managed with more aggressive treatment to avoid irreversible visual loss [126].

The therapeutic approaches to NTG are still strongly debated. Considering that large prospective, multicenter, randomized and controlled clinical trials (i.e., the CNTGS and the EMGT) have demonstrated that an IOP reduction of at least 25–30% from baseline values is effective in delaying the progression of the VF damage in a high percentage of NTG patients [66,67], the current literature strongly supports the pathogenic role of the IOP in NTG. It is thus important to note that despite the fact that individuals with NTG have by definition normal IOP levels, lowering the IOP remains the gold standard in the NTG treatment [30,68,97].

Many different ocular hypotensive drugs are available on the market: prostaglandin and prostamide analogs, beta-blockers, alpha agonists and carbonic anhydrase inhibitors are examples of topical glaucoma drugs that are frequently used to reduce IOP, taken both as topical single therapy or in combination. Prostaglandins and prostamide analogs are the most safe and effective IOP-lowering medications and represent the first choice in NTG therapy [68,71,72,73].

The use of local and systemic beta-blockers and oral calcium channel blockers, especially in the evening, is of particular concern in NTG patients, because they may induce severe nocturnal systemic hypotension, with subsequent ocular perfusion pressure drop [42,71,75,118], which is considered one of the most important risk factors for NTG onset and progression [36,44].

Besides providing an IOP-lowering effect, some ocular hypotensive drugs have also shown the ability to increase the ONH blood flow (latanoprost, bimatoprost, betaxolol, carteolol, levobunolol, carvedilol and nebivolol) or neuroprotective properties (brimonidine, betaxolol, carteolol, carvedilol, latanoprost, bimatoprost and tafluprost) [25,70,72,78,84,85,89]. These adjunctive properties could be particularly useful in treating NTG patients. Moreover, the confirmation of the ability of the novel beta-blockers, carvedilol and nebivolol, to reduce IOP and increase the ocular blood flow in clinical trials in glaucomatous patients may lead to the development of new glaucoma therapies.

IOP-lowering treatment management should be tailored specifically to each patient. Treatment may need to be changed as the disease progresses or a patient’s response to medicine changes. For optimal management of NTG, regular monitoring of IOP, VF and OHN by eye care specialists is important. Moreover, performing diurnal IOP curves and addressing IOP peaks are considered to be the most important therapeutic strategies in NTG patients with normal office IOP values [36,68].

Considering that NTG patients have, by definition, a baseline IOP within the statistically normal range, it is often difficult to reduce the IOP values with medications alone, so that non-medical options are often used, including laser and surgical treatments [68]. The CNTGS [66] showed that 57% of the patients achieved a 30% IOP reduction with topical medications, laser trabeculoplasty or both; the remaining 43% required filtering surgery, which remains the most proven option in the treatment of NTG patients when medications or laser treatments are unable to stop the VF and/or OHN damage progression [68,93]. Nonetheless, this issue extends beyond the aim of the present review. 

Nonetheless, the same large clinical trials supporting the central importance of decreasing IOP in NTG subjects have shown, simultaneously, that the IOP reduction alone is not able to decrease or stop the glaucoma damage progression in all cases [66,67], suggesting that other IOP-independent variables may have a role in the pathogenesis of NTG [23,36].

Several alternative IOP-independent therapeutic approaches have been thus attempted for the NTG treatment [23,24,25,36,41,98,99,100].

Although the exact mechanism or compound is often difficult to identify, many natural compounds or drugs approved for treating other pathologies have demonstrated their utility in treating NTG in vitro by showing RGCs neuroprotection properties, in glaucoma animal models or in clinical studies on NTG patients.

An IOP-lowering effect has been demonstrated using forskolin, marijuana, ginseng, resveratrol and caffeine. CCBs, memantine, Ginkgo biloba, Lycium barbarum, Diospyros kaki, T. wilfordii, curcumin, caffeine, crocin, anthocyanins, coenzyme Q10 and vitamins B3, D and E have shown neuroprotective effects on the RGCs due to antioxidant, anti-inflammatory and anti-apoptosis pathways. C-PAP therapy, Ginkgo biloba, green tea, ginseng, anthocyanins and Licium barbarum have been demonstrated to increase ocular blood flow. Diet implementation with citicoline, coenzyme Q 10, EGCG and PEA has been associated with increased PERG and pattern PEV responses. Finally, Ginkgo biloba, citicoline and anthocyanins have shown some protective effects on the VF indices [41,98,99,100,102,107].

Actually, citicoline seems to be the only dietary supplement showing robust evidence as a neuroprotective agent in glaucomatous and also in neurological patients, and it has therefore been approved by the European Union and the Italian Ministry of Healthy as a dietary food supplement for special medical purposes in glaucoma patients [106].

Although these data suggest a rationale for alternative drugs and dietary supplementation in NTG patients, unfortunately, no hard data are available at the present moment demonstrating that the treatment with IOP-independent strategies will result in better preservation of VF in NTG [24,30,68,69,97,101].

In particular, all clinical trials (though not registration studies) investigating the role of alternative drugs and dietary supplements in NTG patients are affected by a small cohort of participants, heterogeneous study design and short follow-up period, so that the clinical applicability of all these substances as co-adjuvants in the prevention and treatment of NTG remains inconclusive [97,98,99,100,101,102]. Further long-term prospective randomized clinical trials are necessary to determine the therapeutic efficacy and safety of the IOP-independent therapies that, one demonstrated, could become part of the NTG treatment. 

Another important consideration is the safety of nutritional supplementation. Beta-carotene and zinc supplementations have been associated with a higher risk of lung cancer, especially amongst smoking patients; similarly, excess doses of alpha-tocopherol are associated with increased sub-arachnoidal bleeding risk, ascorbic acid is associated with kidney stones and retinol can be associated with raised intracranial pressure [99,127]. 

Although further studies are definitely required, the literature about the use of IOP-independent strategies in NTG may suggest the following points [23,98,99,100,102]:-To avoid nonselective topical beta-blockers in the evening;-To avoid any systemic anti-hypertensive medications at nighttime, because both beta-blockers and calcium channel blockers may have a negative impact on ONH perfusion and oxygenation;-To implement the diet with antioxidants;-To diagnose and possibly treat the systemic disorders typically associated with the NTG, such as systemic hyper- and hypotension, OSAS, hyperlipidemia, hyperglycemia, anemia, congestive heart failure, transient ischemic attacks, cardiac arrhythmias and vitamin deficiencies.

Future IOP-independent therapies for NTG patients could include the following strategies [24,25]:-Neurotrophic factors that are important for neurons growth, differentiation and survival;-Gene therapy that can protect RGCs by transferring some foreign genes;-Stem-cells-based therapy that can integrate and replace dead RGCs.

In conclusion, NTG is a subtype of open-angle glaucoma with unique clinical features, systemic pathologies association and management challenges. The goal of pharmaceutical treatments for NTG is to stop or delay the VF damage progression and to prevent central vision loss. No clinically proven treatments alternative to IOP reduction and control are available to date. The first choice to reduce the IOP is the use of topical ocular hypotensive drugs. When the IOP cannot be efficiently controlled with maximum local topical therapy, in addition to visual field defects and central vision progressive loss, alternative treatment methods like laser therapy (i.e., selective laser trabeculoplasty) and surgery need to be considered. For tracking the evolution of the NTG disease and making necessary adjustments to the treatment plan, adherence to the prescribed course of action and regular follow-up visits with eye care professionals are essential. 

Literature in the field of NTG has shown considerable evidence that suggests that IOP-independent risk factors, such as vascular dysregulation, accelerated apoptosis, inflammation and oxidative stress, may play an important role in the pathogenesis of this disease. Alternative potential therapeutic options that may play a role in these potential risk factors are currently being studied to improve the management of NTG. Unfortunately, clinical studies investigating the role of alternative drugs and dietary supplements to prevent and treat NTG patients remain inconclusive to date. The identification of risk factors other than IOP in NTG patients strongly suggests that clinicians should control additional risk factors, such as systemic hyper- or hypotension, diabetes, anemia and vascular conditions, which may hasten the onset and progression of their glaucoma. Overall, a better prognosis and greater vision preservation can be achieved with early diagnosis, risk factors identification, diligent management and patient compliance. 

The ideal NTG treatment should require a few administrations during the day, have no side effects and provide IOP reduction, increased ONH vascular perfusion and RGCs neuroprotection and regeneration, thus targeting both IOP and IOP-independent risk factors. Further comprehension of the pathophysiology of the NTG will help the clinicians to understand when to use IOP-lowering treatments and when to adopt additionally or alternatively a therapy directed at specific abnormal factors related to the pathogenic process in a specific subject.

## Figures and Tables

**Table 1 pharmaceuticals-16-01172-t001:** Pharmacological effects of the drugs used to reduce the IOP in glaucomatous patients.

Substances		IOP-Lowering Effect			OBF Modification			Neuroprotection		
	Animal	OAG	NTG	Animal	OAG	NTG	In Vitro	Animal	OAG	NTG
	Models	Patients	Patients	Models	Patients	Patients	Models	Models	Patients	Patients
* PGA and prostamide analogs *	D	D	D	D >	D >	D >	D	D		
Latanoprost	D	D	D	D >	D >	D >				
Travoprost	D	D	D	D >	D >	D >				
Bimatoprost	D	D	D	D >	D >	D >	D	D		
Tafluprost	D	D	D	D >	D >	D >	D	D		
* NO-donating PGA analogs *										
Latanoprostene bunod	D	D	D							
* β-adrenergic antagonists *										
Timolol	D	D	D	D <	D <		D	D		
Carteolol	D	D	D	D >	D >		D	D	S	S
Levobunolol	D	D	D	D >	D >		D	D		
Betaxolol	D	D	D	D >	D >		D	D	S	S
Carvedilol	D			D >			D	D		
Nebivolol	D			D >	D >					
* α-2-adrenergic agonists *										
Brimonidine	D	D	D				D	D	S	S
* Carbonic anhydrase inhibitors *										
Dorzolamide	D	D	D	D >	D >	D >	D			
Brinzolamide	D	D	D							
Acetazolamide	D	D	D							
* Miotics *										
Pilocarpine	D	D	D				S			
* ROCK inh. *										
Sovesudil	D	D	D	D >	D >		D	D		
Netarsuldil	D	D	D	D >	D >		D	D		

IOP = intraocular pressure; OBF = ocular blood flow; OAG = open-angle glaucoma; NTG = normal tension glaucoma; D = demonstrated, i.e., supported by strong scientific evidence; S = supposed, i.e., supported by weak scientific evidence; > = increase; < = decrease; PGA = prostaglandin; NO = nitric oxide; ROCK = Rho-associate protein kinase.

**Table 2 pharmaceuticals-16-01172-t002:** Biological effects of the dietary supplements used in glaucomatous patients.

Substances		IOP Changes			OBF Changes			Neuroprotection		
	Animal	OAG	NTG	Animal	OAG	NTG	In Vitro	Animal	OAG	NTG
	Models	Patients	Patients	Models	Patients	Patients	Models	Models	Patients	Patients
Flavonoids				D >	D >		D	D		
Ginkgo biloba extract				D >	D >	D >	D	D	S	S
Camelia sinensis or Green tea				D >			D	D	S	S
Anthocyanins		D <			D >	D >	D	D	S	S
Epigallocatechin-3-gallate							D	D	S	
Resveratrol							D	D		
Citicoline							D	D	D *	S
Coenzyme Q10 (ubiquinone)	D <						D	D	S	
Forskolin	D <	D <					D	D	S	
Panax Ginseng					D >	D >	D	D	S	
Curcumin							D	D		
Cannabinoids	D <	D <					D			
Palmitoylethanolamide (PEA)		D <	D <					D	S	S
Nitric oxide (NO)	D <	D <	D <							
Crocus sativus L. or Saffron		D <					D	D		
Crocin and Crocetin				D >			D	D		
Taurine							D	D		
Lycium barbarum							D	D		
Erigeron Breviscapus Hand. Mazz.	D <						D	S	S	
Hesperidin							D	D		
Scutellaria baicalensis Georgi	D <						D	D		
Diospyros kaki L.	D <						D	D		
Tripterygium wilfordii Hook F.							D	D		
Lutein and Zeaxanthin							D	D		
Caffeine	D <	D </>						D		
Nicotine					D <		D	D		
Melatonin	D <	D <					D	D		
Ethilic alcohol		D </>								
Vitamin A (retinol)									S	
Vitamin B3 (niacine)/nicotinamide							D	D	S	
Vitamin B6 (pyridoxine)							D		S	
Vitamin B12 (cobalamin)							D			S
Vitamin C (ascorbic acid)	D <	D <					D	D	S	
Vitamin D (cholecalciferol)	D <							D		
Vitamin E (alpha-tocopherol)					D >		D	D	S	
Omega-3 PUFAs	D <	D <						D		
Alfa-lipoic acid							D	D		
Apolipoprotein-E							D	D		
Zinc							D	D		
Magnesium				D >			D	D		
Hydrogen sulfide		D <					D			
Creatine							D			
Spermidine							D	D		
Nuclear factor-kappa B							D	D		

IOP = intraocular pressure; OBF = ocular blood flow; OAG = open-angle glaucoma; NTG = normal tension glaucoma; D = demonstrated, i.e., supported by strong scientific evidence; S = supposed, i.e., supported by weak scientific evidence; > = increase; < = decrease; PFA = polyunsaturated fatty acids; * = also demonstrated in patients affected by neurological disorders, such as Parkinson’s and Alzheimer’s diseases, dementia and stroke.

**Table 3 pharmaceuticals-16-01172-t003:** Biological effects of the drugs showing IOP-independent actions in glaucomatous patients.

Substances		IOP-Lowering Effect			Ocular Blood Flow Alteration			Neuro-Protection		
	Animal	OAG	NTG	Animal	OAG	NTG	In Vitro	Animal	OAG	NTG
	Models	Patients	Patients	Models	Patients	Patients	Models	Models	Patients	Patients
Calcium channel blockers				D >/<	D >/<	D >/<	D	D	S	S
Memantine							D	S		
ACEIs	D	D					D			
Anticonvulsants (valproic acid)							D	D		
Edaravone								D		
N-acetylcysteine								D		
Statins									S	S
Androgen deprivation therapy									S	S
Minocycline								D		
Azithromycin							D			
cAMP PI								D		
C-PAP									S	S

IOP = intraocular pressure; OBF = ocular blood flow; OAG = open-angle glaucoma; NTG = normal tension glaucoma; D = demonstrated, i.e., supported by strong scientific evidence; S = supposed, i.e., supported by weak scientific evidence; > = increase; < = decrease; PFA = polyunsaturated fatty acids; ACEIs = angiotensin-converting enzyme inhibitors; cAMP PI = cyclic adenosine monophosphate phosphodiesterase inhibitors; C-PAP = continuous positive airway pressure.

## Data Availability

Not applicable.

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
