# Peer review of "Pharmaceutical Approaches to Normal Tension Glaucoma"

_pharmaceuticals, 2023, doi:10.3390/ph16081172_

Round 1
Reviewer 1 Report
This review contains the conception, pathophysiology,and drugs of normal tension glaucoma. The logic is clear and the content is rich. However, there’re some shortcomings:
1. The definitions of normal tension glaucoma in different countries are different. Authors should present some definitions of normal tension glaucoma according to some important guidelines.
2. In line 420-422, “In more recent years, large clinical trials have demonstrated that the reduction of the IOP is effective in or delaying the progression, and probably preventing the onset, of the VF damage in NTG patients” The word “or” should be deleted.
3.In Line 504 “Latanoprostene bunod” is one of nitric-oxide-donating PGs, there’re some other types of nitric-oxide-donating PG, for example, NCX470. Authors should use a type name instead of lataoprostene bunod.
Author Response
This review contains the conception, pathophysiology, and drugs of normal tension glaucoma. The logic is clear and the content is rich. However, there’re some shortcomings:
We are grateful to the Reviewer for the positive comments regarding our paper.
- The definitions of normal tension glaucoma in different countries are different. Authors should present some definitions of normal tension glaucoma according to some important guidelines.
To address this important issue, the following has been added, with the appropriate references, regarding the definition of NTG in different parts of the world:
“Normal tension glaucoma (NTG), first recognized as a clinical entity by von Graefe in 1857 (4), is a subtype of primary glaucoma characterized by open angle and IOP values constantly within the statistically normal range without treatment (5). The definitions of NTG, however, may vary slightly amongst different countries. The European Glaucoma Society (EGS) Guidelines, published in 2021 (3), state that “NTG is a specific type of POAG characterized by glaucomatous optic nerve head damage and corresponding visual field defects in patients with IOP consistently less than 21 mmHg”. The Preferred Practice Pattern Guidelines published in 2021 by the American Academy of Ophthalmology (AAO) defines the NTG as “a common form of POAG, i.e. a chronic, progressive optic neuropathy that results in a characteristic optic nerve head cupping, retinal nerve fiber layer thinning and functional visual field loss, in which there is no measured elevation of the IOP.” (6).
In 2015, the Canadian Ophthalmological Society Guidelines reported NTG as “a subgroup of POAG with characteristic visual field defects and glaucomatous optic nerve head changes in patients having normal IOP levels less than 21 mmHg.” (7). The Asia-Pacific Glaucoma Guidelines, published in 2016, reported that “the NTG is a condition in which the typical glaucomatous progressive optic nerve damage and visual filed loss occurs although the intraocular pressure remains normal.”(8). The Japanese Glaucoma Society (JGS) published guidelines in 2023, which defined NTG as “a subtype of POAG in which the IOP always remains within the statistically determined normal range during the developmental process of GON.” (9).
- Gedde SJ, Vinod K, Wright MM, Muir KW, Lind JT, Chen PP, Li T, Mansberger SL; American Academy of Ophthalmology Preferred Practice Pattern Glaucoma Panel. Primary Open-Angle Glaucoma Preferred Practice Pattern®. Ophthalmology. 2021 Jan;128(1):P71-P150. doi: 10.1016/j.ophtha.2020.10.022. Epub 2020 Nov 12. PMID: 34933745.
- Wu AM, Wu CM, Young BK, Wu DJ, Chen A, Margo CE, Greenberg PB. Evaluation of primary open-angle glaucoma clinical practice guidelines. Can J Ophthalmol. 2015 Jun;50(3):192-6. doi: 10.1016/j.jcjo.2015.03.005. PMID: 26040218.
- Asia-Pacific Glaucoma Society (APGS). Amsterdam, The Netherlands: Kugler Publications; 3th edition, 2016. Available from: https://www.apglaucomasociety.org/Public/Public/Resources/APGG. aspx. [Last accessed on 2023 Aug 7].
- Kiuchi Y, Inoue T, Shoji N, Nakamura M, Tanito M; Glaucoma Guideline Preparation Committee, Japan Glaucoma Society. The Japan Glaucoma Society guidelines for glaucoma 5th edition. Jpn J Ophthalmol. 2023 Mar;67(2):189-254. doi: 10.1007/s10384-022-00970-9. Epub 2023 Feb 13. PMID: 36780040.
- In line 420-422, “In more recent years, large clinical trials have demonstrated that the reduction of the IOP is effective in or delaying the progression, and probably preventing the onset, of the VF damage in NTG patients” The word “or” should be deleted.
The error has been corrected accordingly.
3.In Line 504 “Latanoprostene bunod” is one of nitric-oxide-donating PGs, there’re some other types of nitric-oxide-donating PG, for example, NCX470. Authors should use a type name instead of latanoprostene bunod.
Based on the suggestion of the Reviewer, the corresponding paragraph has been entitled as “nitric-oxide-donating PGAs analogues” instead of “latanoprostene bunod”.
We are grateful to the Reviewer for the helpful points provide in the review of our manuscript. We hope that all issues have been addressed in an appropriate manner.
Reviewer 2 Report
1- The review article title should encompass pharmaceutical approaches instead of a singular pharmaceutical approach, as it aims to discuss various types of approaches.
2- Selective and nonselective beta-blockers should be distinguished when referring to beta-adrenergic antagonists.
3- The references related to beta-adrenergic blockers need to be updated, particularly with regards to carvedilol, a nonselective beta blocker, in order to reduce intraocular pressure (IOP).
4- The other substances mentioned in the treatment section at line 98 should be placed at the end of all the treatment approaches.
5- The review article lacks information regarding the prognosis of the disease, which should be included before the conclusion.
6- The conclusion of the review article should provide more comprehensive details.
Author Response
The review article title should encompass pharmaceutical approaches instead of a singular pharmaceutical approach, as it aims to discuss various types of approaches.
As suggested by the Reviewer, the title has been modified to read: “Pharmaceutical approaches to normal tension glaucoma”.
2- Selective and nonselective beta-blockers should be distinguished when referring to beta-adrenergic antagonists.
In accordance with the suggestions made by the Reviewer, the section related to the beta-adrenergic antagonists (beta-blockers) has been modified to distinguish the types of beta-blockers. This section has also been updated to include the new beta-blockers carvedilol and nebivolol. The section has been changed to read:
“ - Beta-adrenergic antagonists (beta-blockers) (Timolol, Levobunolol, Carteolol, Betaxolol, Carvedilol, Nebivolol): since the introduction of timolol for glaucoma therapy in 1979, these drugs have been considered as the first line ocular hypotensive therapy for approximately 25 years (78).
Beta-blockers can be divided into non-selective beta-adrenoceptor antagonists, including timolol, carteolol, levobunolol, and carvedilol, and selective beta-1-adrenoceptor antagonists, including betaxolol and nebivolol, which have been demonstrated to induce less side effects on cardiac and pulmonary functions in comparison with non-selective beta-blockers (78,79).
Beta-blockers induce a diurnal IOP reduction of 20-25% by decreasing aqueous humor production from the epithelial cells of the ciliary body (78). Sleep laboratory studies have demonstrated that beta-blockers do not have any IOP-lowering effect during the nocturnal/sleep time (42,71,74,78). Due to a decrease in endogenous circulating catecholamine levels, the aqueous humor production is significantly lower during the night, which may explain the decreased nocturnal efficacy of the beta-blockers (42,71,74,78).
Potential side effects of the beta-blockers include allergic conjunctivitis, keratitis, bronchospasm, vasospasm, bradycardia, systemic systolic and diastolic hypotension, and heart rate reduction (78,79).
Carteolol is a nonselective beta-adrenoceptor blocker with partial agonist activity (80). These eyedrops, in comparison with those of timolol, although showing comparable IOP decreasing efficacy, lack local anesthetic activity, with consequently less ocular surface irritation, and induce less decrease in heart rate or dyspnea, likely due to the partial agonist activity of the carteolol (80).
Levobunolol is a nonselective beta-blocker that has shown an IOP lowering efficacy similar to that of timolol and a concomitant vasodilatory effect on the vascular smooth muscle cells, likely due to the calcium channels block (81).
Betaxolol is a selective beta-1-adrenoceptor antagonist (82). The use of this eyedrop in NTG is controversial. Studies have shown that similar to timolol, betaxolol can induce hypotensive dipping of blood pressure during nighttime, with detrimental effects on the VF damage progression (83). Comparative clinical trials investigating the use of local beta-blocker for managing NTG have shown a higher decrease in IOP with timolol, however, better VF preservation using betaxolol (82), suggesting that the decrease in blood pressure due to the systemic effect of nonselective beta-blockers could have a negative effect on the visual function preservation.
Carvedilol is a relatively new nonselective beta-blocker with multiple other actions, including antioxidant activity, vasodilatation, inhibition of apoptosis, anti-inflammatory activity, calcium channel blocking, and mitochondrial protective effects (84). Studies have shown this drug to reduce IOP after topical and oral administration in vivo in animal models (85,86).
Nebivolol is a novel beta-1-selective adrenoceptor antagonist which has been shown to decrease the IOP after topical and oral administration in animal models (85).
Betaxolol, carteolol, and levobunolol eye drops have been associated with increased ONH blood flow in glaucoma patients (78,80-82). Carvedilol has shown the ability to improve ocular microcirculation in animal models after topical or oral administration by blocking the alfa-adrenergic receptors (85). Nebivolol has been shown to improve the ocular blood flow in rabbits (85) and in glaucoma patients suffering from concomitant arterial hypertension after oral administration (87). The effect of nebivolol on ocular hemodynamics is likely related to its known peripheral vasodilatory effects due to the NO-releasing properties (87).
Beta-blockers have shown neuroprotective properties in vitro and in animal models, which has been proposed to be related to their ability to reduce the amount of glutamate entering and damaging the RGCs and to their calcium channel blocking properties (25,78,84,86).
In general, the use of beta-blockers in NTG is debatable because of their possible negative effect on the ONH blood flow. These drugs have known vasoconstrictive properties. Moreover, the absorption of the topical beta-blocker in the systemic circulation and evening oral dose of beta-blocker have been demonstrated to increase the physiologic nocturnal arterial systolic and diastolic hypotension and to reduce the heart rate and the blood oxygen saturation (42,71,78). Supporting these concerns, previous studies have demonstrated that the treatment of NTG patients with timolol or betaxolol increases the risk of VF deterioration (83).
Recent clinical trials on NTG patients have shown that betaxolol and carteolol had a protective effect on the VF indices which was IOP-independent (88).
In addition, the following paragraph of the “5. Considerations, Future Perspectives, and Conclusions” section has been modified as follows:
“Besides providing an IOP-lowering effect, some ocular hypotensive drugs have also shown the ability to increase the ONH blood flow (latanoprost, bimatoprost, betaxolol, carteolol, levobunolol, carvedilol, nebivolol) or neuroprotective properties (brimonidine, betaxolol, carteolol, carvedilol, latanoprost, bimatoprost, tafluprost)(25,70,72,78,84,86,87,89). These adjunctive properties could be beneficial in treating NTG patients. Moreover, the confirmation of the ability of the novel beta-blockers carvedilol and nebivolol to reduce the IOP and increase the ocular blood flow in clinical trials in glaucomatous patients may lead to the development of new glaucoma therapies.”
3- The references related to beta-adrenergic blockers need to be updated, particularly with regards to carvedilol, a nonselective beta blocker, in order to reduce intraocular pressure (IOP).
The following references have been added to update the bibliography regarding beta-blockers:
- Egan B, Flack J, Patel M, Lombera S. Insights on β-blockers for the treatment of hypertension: A survey of health care practitioners. J Clin Hypertens (Greenwich). 2018 Oct;20(10):1464-1472. doi: 10.1111/jch.13375. Epub 2018 Oct 5. PMID: 30289609; PMCID: PMC6220865.
- Henness S, Swainston Harrison T, Keating GM. Ocular carteolol: a review of its use in the management of glaucoma and ocular hypertension. Drugs Aging. 2007;24(6):509-28. doi: 10.2165/00002512-200724060-00007. PMID: 17571916.
- Dong Y, Ishikawa H, Wu Y, Yoshitomi T. Vasodilatory mechanism of levobunolol on vascular smooth muscle cells. Exp Eye Res. 2007 Jun;84(6):1039-46. doi: 10.1016/j.exer.2007.01.010. Epub 2007 Jan 27. PMID: 17459374.
- Liu B, Liu YJ. Carvedilol Promotes Retinal Ganglion Cell Survival Following Optic Nerve Injury via ASK1-p38 MAPK Pathway. CNS Neurol Disord Drug Targets. 2019;18(9):695-704. doi: 10.2174/1871527318666191002095456. PMID: 31577210.
- Szumny D, Szeląg A. The influence of new beta-adrenolytics nebivolol and carvedilol on intraocular pressure and iris blood flow in rabbits. Graefes Arch Clin Exp Ophthalmol. 2014 Jun;252(6):917-23. doi: 10.1007/s00417-014-2623-5. Epub 2014 Apr 5. PMID: 24705852; PMCID: PMC4035558.
- Hassan DH, Abdelmonem R, Abdellatif MM. Formulation and Characterization of Carvedilol Leciplex for Glaucoma Treatment: In-Vitro, Ex-Vivo and In-Vivo Study. Pharmaceutics. 2018 Oct 21;10(4):197. doi: 10.3390/pharmaceutics10040197. PMID: 30347876; PMCID: PMC6321274.
- Zeitz O, Galambos P, Matthiesen N, Wagenfeld L, Schillinger W, Wiermann A, Richard G, Klemm M. Effects of the systemic beta-adrenoceptor antagonist nebivolol on ocular hemodynamics in glaucoma patients. Med Sci Monit. 2008 May;14(5):CR268-275. PMID: 18443551.
4- The other substances mentioned in the treatment section at line 98 should be placed at the end of all the treatment approaches.
The order of the sections has been changed in accordance with the suggestions made by the Reviewer.
5- The review article lacks information regarding the prognosis of the disease, which should be included before the conclusion and 6- The conclusion of the review article should provide more comprehensive details.
The Reviewer makes a good point. With regards to prognosis and conclusion, the following paragraph has been modified to include considerations about the NTG prognosis. Moreover, as suggested, additional details have been added to enrichen the conclusion section as recommended:
“5. NTG Prognosis, Considerations about the available pharmacological approaches, Future Perspectives and Conclusions
The prognosis of NTG is variable, and depends on factors that include disease severity at diagnosis, effectiveness of treatment, individual risk factors, overall ocular and general health, etc. (5,35,69,124,125).
The results of the CNTGS provide useful information about the natural history of the NTG disease. This study showed that approximately 65% of untreated NTG eyes did not show any VF damage progression over 7 years of follow-up, suggesting that numerous NTG patients can be monitored without treatment, at least initially, or that, alternatively, have been misdiagnosed. 35% of untreated NTG patients showed a disease deterioration with a highly variable rate of VF loss progression (66).
When a medical, laser, or surgical therapy is applied to the NTG patients in order to reduce the IOP of at least 25-30% from baseline values, disease stabilization was achieved in the 88% of patients in the CNTGS (66), and the 55% of cases in the EMGT (67).
The main risk factors associated with the NTG disease progression in both treated and untreated patients have been demonstrated to be female gender, greater variation in diurnal IOP and diastolic blood pressure, presence of disk hemorrhage, greater vertical cup/disc ratio, and migraine at baseline. The factors of age, mean IOP and baseline IOP were not shown to be risk factors for progression. Epidemiology studies have shown that Asians tend to show a slower rate of disease progression (35,69,124,125). On average, the visual field damage progression has been reported to be slower in NTG than in POAG, but with higher inter-patient variability (5,124).
Similar to other types of glaucoma, NTG can progress to irreversible unilateral or bilateral blindness in the worst cases, even despite therapy (126). The cumulative risk to develop legal unilateral blindness in treated NTG patients followed under standard ophthalmic care has been calculated to be 5.8% and 9.9% at 10 years and 20 years, respectively. The risk for bilateral blindness at 10 and 20 years was respectively 0.3% and 1.4% (126). Patients presenting advanced damage at the diagnosis or rapidly progressing VF loss, the so-called “rapid progressors” are most likely to become blind by the disease. It is fundamental that these patients net to be identified and managed with more aggressive treatment to avoid irreversible visual loss (126).
The therapeutic approaches to NTG are still strongly debated. Considering that large prospective, multicenter, randomized, and controlled clinical trials (i.e. the CNTGS and the EMGT) have demonstrated that an IOP reduction of at least 25%-30% from baseline values is effective in delaying the progression of the VF damage in a high percentage of NTG patients (66,67), the current literature strongly supports the pathogenic role of the IOP in NTG. It is thus important to note that despite the fact that individuals with NTG have by definition normal IOP levels, lowering the IOP remains the gold standard in the NTG treatment (30,68,97).
Many different ocular hypotensive drugs are available on the market: prostaglandin and prostamide analogs, beta-blockers, alpha agonists, and carbonic anhydrase inhibitors are examples of topical glaucoma drugs that are frequently used to reduce IOP, taken both as a topical single therapy or in combination. Prostaglandins and prostamide analogs are the most safe and effective IOP-lowering medications and represent the first choice in the NTG therapy (68,71-73).
The use of local and systemic beta-blockers and oral calcium channel blockers, especially in the evening, is of particular concern in NTG patients, because they may induce severe nocturnal systemic hypotension, with subsequent ocular perfusion pressure drop (42,71,78,118), which is considered one of the most important risk factors for NTG onset and progression (36,44).
Besides providing an IOP-lowering effect, some ocular hypotensive drugs have also shown the ability to increase the ONH blood flow (latanoprost, bimatoprost, betaxolol, carteolol, levobunolol, carvedilol, nebivolol) or neuroprotective properties (brimonidine, betaxolol, carteolol, carvedilol, latanoprost, bimatoprost, tafluprost)(25,70,72,78,84,85,89). These adjunctive properties could be beneficial in treating NTG patients. Moreover, the confirmation of the ability of the novel beta-blockers carvedilol and nebivolol to reduce the IOP and increase the ocular blood flow in clinical trials in glaucomatous patients may lead to the development of new glaucoma therapies.
Management for IOP lowering treatment should be tailored specifically to each patient. Treatment may need to be changed as the disease progresses or a patient's response to medicine changes so for optimal management of NTG, regular IOP, VF, and OHN monitoring by eye care specialists are important. Moreover, performing diurnal IOP curves and addressing IOP peaks is considered to be the most important therapeutic strategy in NTG patients having normal office IOP values (36,70).
Considering that NTG patients… with raised intracranial pressure (99,127).
Although further studies are definitely required, the literature about the use of IOP-independent strategies in NTG may suggest the following points (23,98-100,102):
- To avoid nonselective topical beta-blockers in the evening;
- To avoid any systemic anti-hypertensive medications at nighttime, because both beta-blockers and calcium channel blockers may have a negative impact on ONH perfusion and oxygenation
- To implement the diet with antioxidants;
- To diagnose and possibly treat the systemic disorders typically associated with the NTG, such as systemic hyper- and hypo-tension, OSAS, hyperlipidemia, and hyperglycemia. anemia, congestive heart failure, transient ischemic attacks, cardiac arrhythmias, vitamin deficiencies;
Future IOP-independent therapies for NTG patients could include the following strategies (24,25):
- Neurotrophic factors, that are important for neurons growth, differentiation and survival;
- Gene therapy, that can protect RGCs by transferring some foreign genes;
- Stem cells-based therapy, that can integrate and replace dead RGCs.
In conclusion, NTG is a subtype of open-angle glaucoma with unique clinical features, systemic pathologies association, and management challenges. The goal of pharmaceutical treatments for NTG is to stop or delay the VF damage progression and to prevent central vision loss. No clinically proven treatments alternative to IOP reduction and control are available to date. The first choice to reduce the IOP is the use of topical ocular hypotensive drugs. When the IOP cannot be efficiently controlled with maximum local topical therapy, in addition to visual field defects and central vision progressive loss, alternative treatment methods like laser therapy (i.e. selective laser trabeculoplasty) and surgery must be considered. For tracking the evolution of the NTG disease and making necessary adjustments to the treatment plan, adherence to the prescribed course of action and regular follow-up visits with eye care professionals is essential.
Literature in the field of NTG has shown considerable evidence that suggests that IOP-independent risk factors, such as vascular dysregulation, accelerated apoptosis, inflammation, and oxidative stress, may play an important role in the pathogenesis of this disease. Alternative potential therapeutic options that may play a role in these potential risk factors are currently being studied to improve the management of NTG. Unfortunately, clinical studies investigating the role of alternative drugs and dietary supplements to prevent and treat NTG patients remain inconclusive to date. The identification of risk factors other than IOP in NTG patients strongly suggests clinicians to control additional risk factors such as systemic hyper- or hypotension, diabetes, anemia, and vascular conditions that may hasten the onset and progression of their glaucoma. Overall, a better prognosis and enhanced vision preservation can be achieved with early diagnosis, risk factors identification, diligent management, and patient compliance.
The ideal NTG treatment should require few administrations during the day, have no side effects, and provide IOP reduction, increased ONH vascular perfusion and RGCs neuroprotection and regeneration, thus targeting both IOP and IOP-independent risk factors. Further comprehension of the pathophysiology of the NTG will help the clinicians to understand when to use IOP-lowering treatments and when to adopt additionally or alternatively a therapy directed at particular abnormal factors related to the pathogenic process in a specific subject.”
The following references have been added to enrichen this paragraph:
- Bach-Holm D, Jensen PK, Kessing SV. Long-term rate of progression and target intraocular pressure in patients with normal-tension glaucoma in clinical care. Acta Ophthalmol. 2018 Dec;96(8):e1034-e1035. doi: 10.1111/aos.13767. Epub 2018 Sep 26. PMID: 30259686.
- Sakata R, Yoshitomi T, Iwase A, Matsumoto C, Higashide T, Shirakashi M, Aihara M, Sugiyama K, Araie M; Lower Normal Pressure Glaucoma Study Members in Japan Glaucoma Society. Factors Associated with Progression of Japanese Open-Angle Glaucoma with Lower Normal Intraocular Pressure. Ophthalmology. 2019 Aug;126(8):1107-1116. doi: 10.1016/j.ophtha.2018.12.029. Epub 2018 Dec 31. PMID: 30605741.
- Sawada A, Rivera JA, Takagi D, Nishida T, Yamamoto T. Progression to Legal Blindness in Patients With Normal Tension Glaucoma: Hospital-Based Study. Invest Ophthalmol Vis Sci. 2015 Jun;56(6):3635-41. doi: 10.1167/iovs.14-16093. PMID: 26066741.
7.Three tables are required by the Editor.
Three tables have been added to the manuscript, as suggested by the Editor.
Fourteen references have been added to the revised manuscript.
Moreover, the section relative to the dietary supplements used in glaucomatous patients has been enriched in order to include other substances, as shown in Table 2 and in the text as follows:
“Other substances: several other molecules have shown a neuroprotective effect on RGCs and their axons in cell culture and animal models, which include: lutein and zeaxanthin; nitric oxide; flavonoids; crocus sativus L. or saffron; hesperidin; nicotine; ethylic alcohol; crocin and crocetin; zinc; magnesium; curcumin, spermidine; creatine; alfa-lipoic acid; apolipoprotein-E; nuclear factor-kappa B; omega-3 and omega-6 polyunsaturated fatty acids; melatonin; taurine; forskolin; Lycium barbarum; Erigeron breviscapus Hand.Mazz.; Scutellaria baicalensis Georgi; Diospyros kaki L.; Tripterygium wilfordii Hook F; caffeine (23,25,41,98-100,102)”.
We thank the Reviewer for a very thorough evaluation of our manuscript. Excellent points have been raised that can improve our manuscript. Modifications have been made based on the issues raised. We have tried to address each point, which can be viewed with the track changes throughout the manuscript.